# Generation and propagation of stick-slip waves over a fault with rate-independent friction

Iuliia Karachevtseva[1], Arcady V. Dyskin[2] and Elena Pasternak[1]

[1]School of Mechanical and Chemical Engineering, The University of Western Australia, Australia
[2]School of Civil and Resource Engineering, The University of Western Australia, Australia

*Correspondence to:* Iuliia Karachevtseva (juliso22@gmail.com)

**Abstract.** Stick-slip sliding is observed at various scales in fault sliding and the accompanied seismic events. It is conventionally assumed that the mechanism of stick-slip over geomaterials lies in the rate dependence of friction. However, the movement resembling the stick-slip could be associated with elastic oscillations of the rock around the fault, which occurs irrespective of the rate properties of the friction. In order to investigate this mechanism, two simple models are considered in this paper: a mass-spring model of self-maintaining oscillations and a one-dimensional (1D) model of wave propagation through an infinite elastic rod. The rod slides with friction over a stiff base. The sliding is resisted by elastic shear springs. The results show that the frictional sliding in the mass-spring model generates oscillations that resemble the stick-slip motion. Furthermore, it was observed that the stick-slip-like motion occurs even when the frictional coefficient is constant. The 1D wave propagation model predicts that despite the presence of shear springs the frictional sliding waves move with the p-wave velocity, denoting the wave as intersonic. It was also observed that the amplitude of sliding is decreased with time. This effect might provide an explanation to the observed intersonic rupture propagation over faults.

## 1 Introduction

Earthquakes can lead to catastrophic structural failures and may trigger tsunamis, landslides and volcanic activities (Ghobarah et al., 2006; Bird and Bommer, 2004). The earthquakes are generated at faults, and are either produced by rapid (sometimes 'supersonic') propagation of shear cracks/ruptures along the faults, or originated in the stick-slip sliding over the fault. The velocity of rupture propagation is crucial for estimating the earthquake damage. The rupture velocities can be classified by comparison its speed with the speeds of stress waves in the rupturing solid (Rosakis, 2002). There are several types of rupture propagation: supersonic ($V>V_P$), intersonic ($V_S<V<V_P$), subsonic ($V<V_S$), supershear ($V>V_S$), sub-shear ($V_R<V<V_S$) and sub-Rayleigh ($V<V_R$). According to the data obtained from the seismic observation of crustal earthquakes, most ruptures propagate with an average velocity that is about 80% of the shear wave velocity (Heaton, 1990). However, in some cases, supershear propagation of earthquake-generating shear ruptures or sliding is observed (Archuleta, 1984; Bouchon et al., 2000, 2001, 2010; Dunham and Archuleta, 2004; Aagaard and Heaton, 2004). The above observations introduced the concept of supershear crack propagation (e.g., Bizzarri and Spudich, 2008; Lu at al., 2009; Bhat et al., 2007; Dunham, 2007). However, due to the lack of strong motion recording, there are still some debates regarding the data interpretation (Delouis et al., 2002; Bhat et al., 2007). For instance, it was suggested that the 2002 Denali Earthquake was propagated at a supershear speed of about 40 km (Dunham and Archuleta, 2004). However, the data was based on a single ground motion record. The joint inversion of the combined data-set provides a more robust description of the rupture. The recent studies, which are aimed at deriving the kinematic models for large earthquakes, have shown the importance of the type of data used. It has been shown that slip maps for a given earthquakes may vary significantly (Cotton and Campillo, 1995; Cohee and Beroza, 1994a).

The analytical (e.g., Burridge, 1973) and numerical (e.g., Das and Aki, 1977) research in fracture dynamics indicate that only the Mode II rupture (shear-induced slip occurring in the direction perpendicular to the crack front) can propagate with intersonic velocity ($V_s<V<V_p$) for short durations, as long as the prestress of the fault is high compared to both failure and residual stresses (Dunham, 2007). Intersonic Mode II crack propagation was first confirmed in laboratory by Rosakis et al. (1999).

Sliding over pre-existing fractures and interfaces is one of the forms of instability in geomaterials. It is often accompanied by stick-slip – a spontaneous jerking motion between two contacting bodies sliding over each over. It is assumed that the mechanism of stick-slip lies in intermittent change between static and kinetic friction and the rate dependence of the frictional coefficient (Popp and Rudolph, 2004).

The investigation of the friction law on geological faults is the key element in the modelling of earthquakes. Rate- and state-dependent friction laws proposed by Dieterich, Ruina and Rice (Dieterich, 1978; Ruina, 1983; Rice, 1983) have successfully modelled frictional sliding and earthquake phenomena. There are two types of frictional sliding between surfaces that include the tectonic plates. The first type occurs when two surfaces slip steadily ($V=V_0$ condition, where $V$ - is relative velocity, $V_0$ - is the load point velocity) and is analogous to the fault creep (Byerlee and Summers, 1975). In the stable state, the sliding over discontinuities (faults and fractures) is prevented by friction. Modelling of the frictional sliding is an important tool for understanding the initiation and the development of rupture, and also, the healing of the faults. Many models and numerical methods are developed to describe seismic activities and the supershear fracture/rupture propagation (Noda and Lapusta, 2013; Lapusta and Rice, 2003; Lu at al., 2009; Lapusta et al., 2000; Sobolev, 2011; Bak and Tang, 1989; Harris and Day, 1993).

The faults are continuously subjected to variations in both shear and normal stresses, and can produce sliding over initially stable fractures or interfaces (Boettcher and Marone, 2004). In the Earth's crust, the increase in shear stress is an obvious consequence of tectonic movement, while oscillations in the normal stress can be associated with the tidal stresses or seismic waves generated by other seismic events. These can generate the second dynamic state when the sliding occurs jerkily (slip, stick and then slip again). This type of sliding is called "stick-slip" sliding which exhibit cyclic behaviour. Brace and Byerlee supposed that the stick-slip instabilities in the tectonic plates are associated with the appearance of earthquakes (Brace and Byerlee, 1966). Both types of sliding are usually investigated using a spring-block model introduced by Burridge and Knopoff in 1967 (Turcotte, 1992). The BK model consists of an assembly of blocks, where each block is connected via the elastic springs to the next block and to the moving plate.

In the present paper, we firstly simulate a single element block model, which is one block undergoing frictional sliding on a stiff base. The movement is caused by a spring attached to the block. The other end of the spring moves with a constant velocity. The paper begins with considering stick-slip-like movement occurring under rate-independent friction due to the eigen oscillations of the fault faces and the associated wave propagation. This demonstrates that the rate dependence of friction is not necessarily a controlling phenomenon. We also analyse a simple mechanism of unusually high shear fracture or sliding zone propagation, also referred as the p-sonic propagation of sliding area over a frictional fault. The analysis is based on the fact that accumulation of elastic energy in the sliding plates on both sides of the fault can produce oscillations in the velocity of sliding even if the frictional coefficient is constant. We note that Walker and Shearer (2009) found evidence of the intersonic rupture speeds close to the local P-wave velocity by analysing the Kokoxili and Denali earthquakes seismic data. This paper considers a highly simplified 1-D rod model where many properties of real fault system have been neglected. (Considerable fault geometry simplification is in use in analysing intersonic ruptures, e.g., Bouchon et al., 2010.)

**2 Single degree of freedom frictional oscillator**
We start with the self-excited oscillations, which resembles the stick-slip-like motion, but occurring under constant friction.
A single degree of freedom block-spring model is used for this purpose. A block sliding on a rigid horizontal surface is
driven by a spring whose other end is attached to a driver moving with a constant velocity (Figure 1). All variables and
constants used in equations are listed below in Table 1.

**Table 1: The list of variables and constants**

| Symbol | Meaning | Symbol | Meaning |
|---|---|---|---|
| $V_0$ | load point velocity | $\tau$ | shear stress |
| $V$ | relative velocity of block | $\tau_f$ | friction stress |
| $k_1$ | single spring stiffness | $E$ | Young's modulus |
| $m$ | block mass | $c$ | velocity of longitudinal wave (p=wave) |
| $N$ | gravity force | $\omega$ | eigen frequency |
| $T$ | shear force | $k_2$ | the spring stiffness relating stress and displacement discontinuity (the difference between the rod displacement and the zero displacement of the base) |
| $\mu$ | friction coefficient | $J_0$ | Bessel function of order 0 |
| $\omega_0$ | eigen frequency | $J_0{}'$ | derivative of Bessel function |
| $t$ | time | $i$ | imaginary unit |
| $h$ | thickness of an infinite rod | $\xi$ | independent variable |
| $\rho$ | volumetric rod density | $z$ | integration variable |
| $\sigma_N$ | uniform compressive load | $f, g$ | arbitrary functions |
| $\sigma$ | longitudinal stress | | |


Friction is assumed to be cohesionless: $T_{cr}=\mu N$, where $T_{cr}$ is the force at which sliding starts.

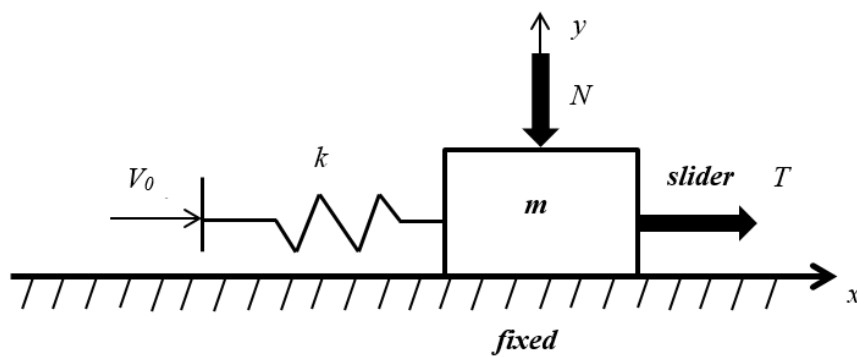

**Figure 1: The single block model.**
The system of equations representing the motion of the block reads:

$$\begin{cases} m\dot{V} = f(T, \mu N) \\ \dot{T} = k_1(V_0 - V) \end{cases}$$    (1)
The appearance of the $f(T, \mu N)$ function in the system of equations represents the fact that $V \geq 0$ .
The function $f(T, \mu N)$ is defined as:
$$f(T, \mu N) = \begin{cases} T - \mu N, \ T > \mu N \text{ and } V > 0 \\ 0, \ T < \mu N \text{ or } V < 0 \end{cases} \qquad (2)$$

In order to represent the system of equations (1) in dimensionless form, it is convenient to introduce a dimensionless time $t^*$:
$$t^* = t\omega_0, \quad \omega_0^2 = \frac{k_1}{m} \qquad (3)$$

where $\omega_0$ is the eigen frequency of the block-spring system, $m$ is the block mass and $k_1$ is the spring stiffness.
The governing system of equations in dimensionless form is defined as:
$$\begin{cases} \dot{V} = f(T^*, \mu N^*) \\ \dot{T} = 1 - V^* \end{cases} \qquad (4)$$

where the dot represents the derivative with respect to dimensionless time $t^*$, and $V^*$, $T^*$ and $N^*$ are the dimensionless
velocity, shear force and gravity force respectively.
$$V^* = \frac{V}{V_0}, \ T^* = \frac{T}{mV_0\omega_0} \ N^* = \frac{N}{mV_0\omega_0}.$$

### 107 2.1 Behaviour of the system

In order to demonstrate the behaviour of the system at stick-slip-type regime, we consider the block sliding under the
following set of initial conditions:
$V(0) = 0, \ \dot{T}(0) = 0$                                                         (5)
Figure 2 represents the corresponding behaviour of the system (dimensinless velocity vs. dimensionless time).

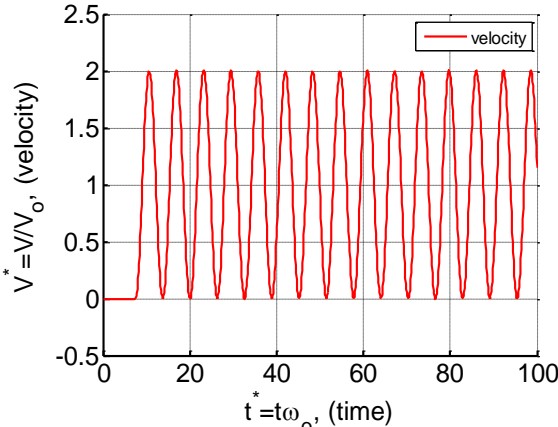

**Figure 2: Block sliding with constant friction coefficient.**
It is observed that the system exhibits self-excited oscillations even with constant friction coefficient, which somewhat
resemble the stick-slip-type sliding. Furthermore, the energy in the system does not change with time, obviously due to the
constant energy influx by velocity $V_0$, where the excess of the $V_0$ is dissipated by friction.
A detailed investigation of the behaviour of a system described in a section 2 was undertaken in our previous works
(Karachevtseva et al., 2014; Karachevtseva et al., 2014). ). It should also be noted that similar oscillation-type movements
were observed in laboratory experiments on sliding of two granite blocks under biaxial compression (Sobolev et al., 2016).

### 122 3 Stress wave propagation in frictional sliding (generalisation 1D solid)

In the previous section, we showed the stick-slip-like motion occurring even when the friction coefficient is constant. In this
section we will expand our understanding to incorporate the slide over a fault where a stick-slip phenomenon is traditionally
flagged as a mechanism of earthquakes. We shall keep assuming the constant friction law, which will permit us to obtain an
analytical solution.  For this purpose, following Nikitin (1998), we consider the simplest possible 1D model of fault sliding,
which takes into account the rock elastic response and the associated dynamic behaviour. The model is shown in Figure 3. It
consists of an infinite elastic rod of height (thickness) $h$, and of unit length in the direction normal to the plane of drawing in
Figure 3. The linear density is $\rho$ and the rod is assumed to be able slide over a stiff surface. The sliding is resisted by
friction. The stiff surface can be described as a symmetry line such that instead of the (horizontal) fault, only the upper half
of the line is considered. The rod is connected to a stiff layer moving with a constant velocity $V_0$. The connection is achieved
through a series of elastic shear springs. Both the elastic rod and the elastic springs describe the model of the elasticity of the
rock around the fault, as shown in Figure 3. We assume that the system is subjected to a uniform compressive load $\sigma_N$ such
that the friction stress is kept constant, which is  assumed equal to $\tau_f = \mu\sigma_N = const$ .

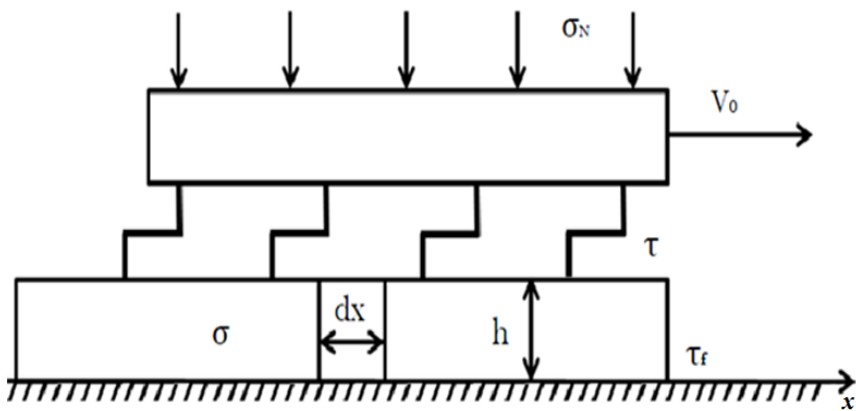


**Figure 3: The model of infinitive elastic rod driven by elastic shear spring.**
Equation of movement of the rod reads:
$$\frac{\partial\sigma}{\partial x}+\frac{1}{h}(\tau-\tau_f)=\rho\frac{\partial V}{\partial t} \tag{6}$$

Where $\sigma$ is the longitudinal (normal) stress in the rod, $\tau$ is the contact shear stress, $\tau_f$ is the frictional stress, $V_0$ is the load
point velocity and $V(x,t)$ is the velocity of point $x$ of the rod at time $t$, as shown in Figure 3.
According to the Hooke's law:
$$\sigma = E\frac{\partial u}{\partial x} \tag{7}$$

where $u(x,t)$ is the displacement and $E$ is the Young's modulus of the rod. After differentiating, we have:
$$\frac{\partial\sigma}{\partial t} = E\frac{\partial V}{\partial x} \tag{8}$$

The elastic reaction of the shear springs is expressed as:
$$\frac{\partial\tau}{\partial t} = k_2(V-V_0) \tag{9}$$

where $k_2$ is the spring stiffness relating stress and displacement discontinuity (the difference between the rod displacement
and the zero displacement of the base).
Defining $\Delta V = V - V_0$ and solving the system of equations (6)-(9), we get the following wave equation:
$$\frac{\partial^2 \Delta V}{\partial t^2} = c^2 \frac{\partial^2 \Delta V}{\partial x^2} - \omega^2 \Delta V \tag{10}$$
where $c = \sqrt{Eh/\rho}$ is the velocity of the longitudinal wave (p-wave), $\omega = \sqrt{k_2/(h\rho)}$ is regarded as eigen frequency of
the system consisting as a unit length of the rod considered as a lamp mass on the shear springs.
It is observed that despite the presence of shear springs and friction between the rod and the stiff surface, the waves
propagate with the p-wave velocity determined by the Young's modulus and density of the rod. Therefore, according to the
terminology described in the introduction, the wave should be named *p-sonic wave*. It should be highlighted that while such
waves look like the shear waves, they are in fact compressive waves propagation along the rod, hence denoted as the p-wave
velocity.
In order to analyse the way the pulse propagates, equation (10) is complemented by the initial conditions as:
$$\Delta V(x,t) = f_0(x); \quad \frac{d\Delta V}{dt} = F_0(x) \tag{11}$$

Solution of wave equation (10) can be found by using the Riemann method (e.g., Koshlyakov, 1964).
$$\Delta V(x,t) = \frac{1}{2}[f(x-ct) + g(x+ct)] + \frac{1}{2} \int_{x-ct}^{x+ct} \Phi(x,t,z)dz \tag{12}$$
where
$$\Phi(x,t,z) = \frac{1}{\sqrt{c^2 t^2 - (z-x)^2}} \varphi(x,t,z) \tag{13}$$
The integral from (12) can be found by using the Chebyshev-Gauss method
$$I(x,t) = \int_{x-ct}^{x+ct} \Phi(x,t,z)dz \approx \frac{\pi}{n} \sum_{j=1}^{n} \varphi(x,t,\ x+\zeta_j at), \quad \xi_j = \cos\left(\frac{2j-1}{2n}\pi\right) \tag{14}$$
where
$$\varphi(x,t,z) = \frac{1}{c} F(z) J_0\left(\frac{\omega}{c} i\sqrt{c^2 t^2 - (z-x)^2}\right) \sqrt{c^2 t^2 - (z-x)^2} + \omega t f(z)\left(\frac{1}{i}\right) J_0{}'\left(i\frac{\omega}{c}\sqrt{c^2 t^2 - (z-x)^2}\right) \tag{15}$$

**3.1 Propagation of an initial sliding**
Figures 3-4 represent the propagation of initial sliding under the different initial conditions. Particularly, a triangular velocity
impulse, equation (16) and zero acceleration were used as initial conditions for Figure 3. As shown in Figure 4, linear and
harmonic functions are used for velocity and acceleration as initial conditions.
$$f(x; a,b,c) = \max\left(\min\left(\frac{x-a}{b-a}, \frac{c-x}{c-b}\right), 0\right) \tag{16}$$
where $x$ is the vector, $a$, $b$, $c$ are scalar parameters.

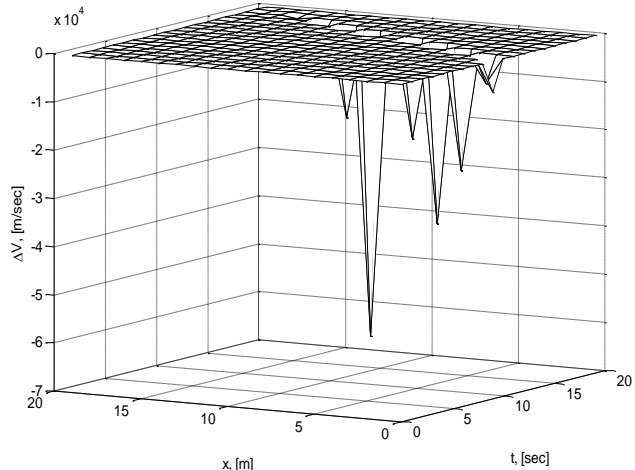


177                Figure 3: Propagation of initial sliding in the form of a triangular function *f*(*z*) of zero area.

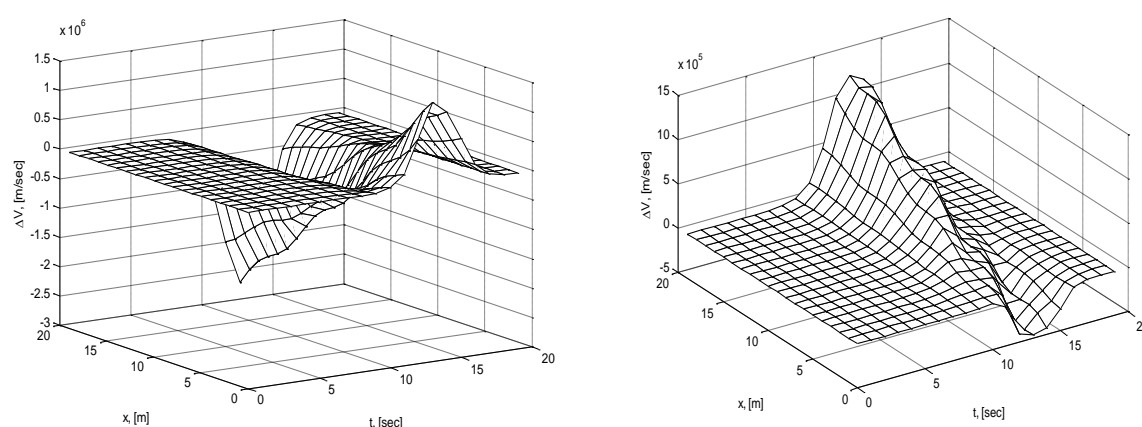


179                        Figure 4: Propagation of initial sliding with different initial conditions.

It is seen that the initial sliding (impulse) propagating with p-wave velocity keeps its width but the amplitude reduces with
time. It is also observed that as the impulse propagates, it loses energy which goes to increase the energy of shear springs.
**4 Discussion**
This paper introduced the notion that the frictional movement resembling the stick-slip sliding, which are often observed and
usually attributed to the rate dependence of friction, can be obtained with constant friction by taking into account the
elasticity of the surrounding and its self-oscillations. This understanding is applied to propagation of slip over infinitely long
fault leads to a simple model that predicts that the slip will propagate with p-wave velocity. This conclusion is made under
the assumption of constant (rate-independent) friction. Relaxing this assumption, that is taking into account that
$\tau_f = \tau_f(\dfrac{\partial \Delta V}{\partial t})$ leads to the following equation replacing equation (10):
$$\left(1+\frac{1}{\rho h}\frac{d\tau_f}{d\Delta V_t'}\right)\frac{\partial^2 \Delta V}{\partial t^2} = c^2 \frac{\partial^2 \Delta V}{\partial x^2} - \omega^2 \Delta V, \quad \Delta V' = \frac{\partial \Delta V}{\partial t} \tag{17}$$
It is seen that when the sliding rate changes slowly, the propagation speed of rupture $c_1$ can be approximated as:
$$c_1{}^2 \approx c^2 \left(1 + \frac{1}{\rho h}\frac{d\tau_f}{d\Delta V_t'}\right)^{-1}$$ (18)
Furthermore, it is observed that when the friction increases with the sliding rate, $c_1$ becomes smaller than p-wave velocity. If
the rate dependence of friction is lowered further, the slip propagation can become intersonic.

## 5 Conclusions

In this paper, it is shown that the accumulation of elastic energy in the sliding plates on both sides of the fault can produce
oscillations in the velocity of sliding even when the friction is constant. These oscillations resemble stick-slip movement, but
they manifest themselves in terms of sliding velocity rather than displacement. The sliding exhibits wave-like propagation
over long faults. Furthermore, the 1D model shows that the zones of sliding propagate along the fault with the velocity of p-
wave (the propagation speed can however be lower if the rate dependence of friction is taken into account). The mechanism
of such fast wave propagation is the normal (tensile/compressive) stresses in the neighbouring elements (normal stresses on
the planes normal to the fault surface) causing a p-wave propagating along the fault rather than the shear stress controlling
the sliding. This manifests itself as a p-sonic propagation of an apparent shear rupture.

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
