# Peer review of "Generation and propagation of stick-slip waves over a fault with"

_Nonlinear Processes in Geophysics, 2016_

## Referee Comment (RC1) · Anonymous Referee #1 · 14 Feb 2017

The presented work is devoted to the vital problem of active fault dynamics. Discussed are the questions of stick-slip emergence under constant friction factor and relations of propagations of disturbances in the medium. The authors have performed numerical experiments on the basis of 1D spring-block Burridge-Knopoff model and have computed the regularities of propagation of disturbances in an infinitely long elastic rod exposed to shear load with a system of elastic springs. It is shown in the work that sliding at alternating velocity may emerge under constant friction factor. It is also shown, that in the frames of the 1D model of infinitely long elastic rod any disturbance propagates at the velocity of p-waves, and as it propagates its amplitude decreases. The assumption is put forward in the work that this effect can account for the supersonic

propagation of rupture along a fault.

I think that using such simple models has to be accurately grounded, and even more arguments are needed to apply the obtained results to real processes taking place in natural fault zones. The oscillations considered in the first part of the work can't be called "stick-slip", because the latter is characterized by a prolonged stage of rest followed by an abrupt sliding.

I offer several comments that I hope will help improving the paper: General comments: 1. There is no "discussion" in the work, where it would be appropriate to discuss in detail the non-linear effects of disturbance propagation obtained in the work and their links to the processes in nature. 2. In Parts 2 and 3 all the variables and constants used in equations should better be listed once in a single table instead of repeating the terms in different equations with different meanings. 3. In Part 3 the simplest 1D case is considered, so, a disturbance, once emerged, can propagate only along the rod, and the law of its propagation is defined by the parameters E and p which means that the disturbance can only propagate at the velocity of p-wave, because no other motion is possible. 4. The captions should be revised to make them more substantial, clarifying and informative.

I think that the paper needs an essential improvement.

Please also note the supplement to this comment:
http://www.nonlin-processes-geophys-discuss.net/npg-2016-82/npg-2016-82-RC1-supplement.pdf

**Supplement:**

**REVIEW**

The presented work is devoted to the vital problem of active fault dynamics. Discussed are the questions of stick-slip emergence under constant friction factor and relations of propagations of disturbances in the medium. The authors have performed numerical experiments on the basis of 1D spring-block Burridge-Knopoff model and have computed the regularities of propagation of disturbances in an infinitely long elastic rod exposed to shear load with a system of elastic springs. It is shown in the work that sliding at alternating velocity may emerge under constant friction factor. It is also shown, that in the frames of the 1D model of infinitely long elastic rod any disturbance propagates at the velocity of p-waves, and as it propagates its amplitude decreases. The assumption is put forward in the work that this effect can account for the supersonic propagation of rupture along a fault.

I think that using such simple models has to be accurately grounded, and even more arguments are needed to apply the obtained results to real processes taking place in natural fault zones. The oscillations considered in the first part of the work can't be called "stick-slip", because the latter is characterized by a prolonged stage of rest followed by an abrupt sliding.

There are some remarks and comments as follows:

1. There is no "discussion" in the work, where it would be appropriate to discuss in detail the non-linear effects of disturbance propagation obtained in the work and their links to the processes in nature.
2. In Parts 2 and 3 all the variables and constants used in equations should better be listed once in a single table instead of repeating the terms in different equations with different meanings.
3. In Part 3 the simplest 1D case is considered, so, a disturbance, once emerged, can propagate only along the rod, and the law of its propagation is defined by the parameters $E$ and $\rho$, which means that the disturbance can only propagate at the velocity of p-wave, because no other motion is possible.
4. The captions should be revised to make them more substantial, clarifying and informative.

Less important remarks:
Raw 38. Cohee and Beroza, 1994a $\rightarrow$ Cohee and Beroza, 1994
Raws 48-49. "However, the faults ... can produce sliding over initially stable fractures/interfaces" – a citation is needed.
Raw 64. The citations should better be replaced by (Brace & Byerlee, 1966).
Raw 82, Eq. 2. As a matter of fact, this equation defines the rule of the frictional force action. When $V=0$ the frictional force can act on a body only provided that the shear force is not zero. In the presented system this condition is not true.
Raw 87, Eq.4. It is faulty to designate dimensional and dimensionless quantities by the same variables. Please, submit relations for all the dimensionless variables.
Raw 93, Eq.5. If all the variables are dimensionless, it is unclear, why the relation $\mu N$ appears? It misses in the plots presented in Fig.2.

Raw 95, Fig.2. Under the action of a frictional force constant modulo, the energy should dissipate, but it doesn't. This fact should be explained.

Raw 105. Fig.2 presents harmonic oscillations, but not the regime of "stick-slip".

Raw 114. $\tau_{fr} = k\mu\sigma_N$. What is k?

Raw 115, Fig.3. There is $\tau_f$ in the figure, but not $\tau_{fr}$.

Raw 126, Eq.8. It is unclear, what is k – the stiffness of a single spring, of all the springs, or the specific stiffness of springs per unit length? Attention should be paid to Eq.1, where the same notation is used.

Raw 129, Eq.9. The formula is presented in a faulty way. If one supposes that $\Delta V=u$ is a re-introduced new value, it appears that the increment of velocity equals to displacement, which is impossible.

Raws 137-145. Equations 11-14. All the constants and variables should be clarified.

Raw 145. Eq.14. What is the function $J_0$, what are the coefficients $i$ и $b$, and what is the difference between the Bessel functions $J_0$ и $J_0'$?

Part 3.1. Since the results are presented in the form of time series of dimensional variables, parameters of the model should be designated, which were used in calculations. The visual presentation of results is not pictorial enough. To my mind, the grid is too coarse. The dimensionality of Y-axis is not mentioned.

Raw 152. Fig.3 (right). It is better to plot all the curves using a single X-axis, and one and the same scale of the Y-axes (may be, it's better to use the logarithmic scale).

Raw 152. Fig.3 (left). Propagation of the disturbance is not seen at all. The Y-axis should be inverted, or even better, re-calculated for the disturbance when u(t, x) > 0.

The function of pulse shape is specified in a poorly comprehensible way. It's better to give it in a standard mathematical form.

Raw 155, Fig.4 (left). The disturbance is not seen in the area of big t. The viewing angle should be changed. No need in the inscriptions in the plot.

Raw 162, Fig.5. The amplitude of the disturbance is maximal at the initial moment and reduces with time (raw 158). But, in the figure the amplitude is zero in the range of 0-9 s, then it increases in the range of 10-14 s, and then it decreases. What really shown in the figure?

In view of all the remarks above, I think that the paper needs an essential improvement.

---

## Referee Comment (RC2) · Anonymous Referee #2 · 27 Feb 2017

The present manuscript reports results of numerical study of a fault dynamics under elastic oscillations of the rocks surrounding the fault under constant friction factor with application of simple 1D spring-block models.

I believe that this work can be made acceptable for publication in NPG, but some major revisions should be made, that will drastically improve the readability of the work and its overall impact.

1) The paper does a poor job of placing the work in a context with previous work that relates fault slip behavior to elastic oscillations of the rock surrounding the fault. Addressing this comment will make the paper more readable to a wide earth science audience and place it in better context to other work that has been done on a similar

**topic.**

2) An application of simple models like the Burridge-Knopoff model and 1D model of an infinite elastic rod driven by elastic shear spring for the declared purpose should be substantiated in details.

3) The constant friction factor used in the models instead of generally accepted rateand-state friction law has to be grounded and supported by lab results and field observations.

4) A discussion section of the manuscript is required for an analysis and comparison of the numerical results and drawn conclusions with published data obtained under laboratory and natural conditions.

Moreover, I realized that the English writing is not good enough, some parts of the text are difficult for understanding, there are some syntax and spelling errors, and I strongly recommend reviewing the text by a native English speaker.

---

## Author Response (AR2)

**Summary of changes**

**(Reviewer #1)**

First, we would like to thank the reviewer for his interest in our work and for helpful comments that will drastically improve the paper. As indicated below, we have checked all comments provided by the reviewer and have addressed necessary changes accordingly to his feedback.

**Below are reviewer's comments and our responses:**

*C1: "There is no "discussion" in the work, where it would be appropriate to discuss in detail the non-linear effects of disturbance propagation obtained in the work and their links to the processes in nature".*

R1: The discussion part has been added into the paper. There the non-linear effects are discussed.

*C2: In Parts 2 and 3 all the variables and constants used in equations should better be listed once in a single table instead of repeating the terms in different equations with different meanings.*

R2: We would like to thank the reviewer for this comment. The variables are now listed in Table 1.

*C3: In Part 3 the simplest 1D case is considered, so, a disturbance, once emerged, can propagate only along the rod, and the law of its propagation is defined by the parameters E and ρ, which means that the disturbance can only propagate at the velocity of p-wave, because no other motion is possible.*

A3: Yes, it is a 1D case, but the shear motion is allowed as well. So, it is not immediately obvious why it should be just p-wave velocity. To emphasise the point we modified the first sentence in the para after (9), which now reads "It is seen that despite the presence of shear springs and friction between the rod and the stiff surface the waves propagate with the p-wave velocity determined by the Young's modulus and density of the rod."

*C4: The captions should be revised to make them more substantial, clarifying and informative.*

A4: Thank you for your comment. It has been done.

**Less important remarks:**

*C5: "Raw 38. Cohee and Beroza, 1994a → Cohee and Beroza, 1994"*

A5: Thank you. It has been done.

*C6: "Raws 48-49. "However, the faults … can produce sliding over initially stable fractures/interfaces" – a citation is needed".*

A6: Thank you. It has been done

*C7: "Raw 64. The citations should better be replaced by (Brace & Byerlee, 1966)".*

A7: Thank you.  It has been done.

*C8: "Raw 82, Eq. 2. As a matter of fact, this equation defines the rule of the frictional force action. When V=0 the frictional force can act on a body only provided that the shear force is not zero. In the presented system this condition is not true".*

A8: We agree with the reviewer; it was a misprint. The system of equations has been corrected.

*C9: "Raw 93, Eq.5. If all the variables are dimensionless, it is unclear, why the relation μN appears? It misses in the plots presented in Fig.2".*

A9: Thank you. The Fig. 2 has been replaced.

*C10: "Raw 95, Fig.2. Under the action of a frictional force constant modulo, the energy should dissipate, but it doesn't. This fact should be explained".*

A10: Thank you. This has been added into the paper. Please see below.
 "The energy in the system does not change with time, obviously due to the constant energy influx by velocity $V_0$ whose excess is dissipated by friction".

*C11: Raw 105. Fig.2 presents harmonic oscillations, but not the regime of "stick-slip".*

A11: These oscillations resemble stick-slip movement, but they manifest themselves in terms of sliding velocity rather than displacement.

*C12: "Raw 114. $\tau fr = k\mu\sigma N$. What is k"?*

A12: Wrong formula was used. It has been corrected.

*C13: "Raw 115, Fig.3. There is $\tau f$ in the figure, but not $\tau fr$".*

A13: Typo was in Eq.6. It has been corrected.

*C14: "Raw 126, Eq.6. It is unclear, what is k – the stiffness of a single spring, of all the springs, or the specific stiffness of springs per unit length? Attention should be paid to Eq.1, where the same notation is used".*

A14: We agree with the reviewer. It has been changed. The details are in the table 1.

*C15: "Raw 129, Eq.9. The formula is presented in a faulty way. If one supposes that ΔV=u is a re-introduced new value, it appears that the increment of velocity equals to displacement, which is impossible".*

A15: Awkward notation was used. *U* was not to be displacement. It has been changed.

*C16: "Raws 137-145. Equations 11-14. All the constants and variables should be clarified".*

A16: It has been done. Please see table 1.

*C17: "Raw 145. Eq.14. What is the function J0, what are the coefficients i u b, and what is the difference between the Bessel functions J0 and J0'''"?*

A17: *i* – is imaginary unit; $J_0$ – is Bessel function; $J_0'$ – is derivative of Bessel function.  Please see table 1.

C18: *"Part 3.1. Since the results are presented in the form of time series of dimensional variables, parameters of the model should be designated, which were used in calculations. The visual presentation of results is not pictorial enough. To my mind, the grid is too coarse. The dimensionality of Y-axis is not mentioned".*

A18: Thank you for suggestion. We have modified the paper structure and data presentation.

C19: *"Raw 152. Fig.3 (right). It is better to plot all the curves using a single X-axis, and one and the same scale of the Y-axes (may be, it's better to use the logarithmic scale)".*

A19: Thank you for your suggestion. A confusing figure was used. It has been deleted.

C20: *"Raw 152. Fig.3 (left). Propagation of the disturbance is not seen at all. The Y-axis should be inverted, or even better, re-calculated for the disturbance when u(t, x) > 0.*
*The function of pulse shape is specified in a poorly comprehensible way. It's better to give it in a standard mathematical form".*

A20: Thank you for your suggestion. The Fig.3 was corrected. A standard mathematical formula was added, please see equation 15.

C21: *"Raw 155, Fig.4 (left). The disturbance is not seen in the area of big t. The viewing angle should be changed. No need in the inscriptions in the plot".*

A21: It has been done.

C22: *"Raw 162, Fig.5. The amplitude of the disturbance is maximal at the initial moment and reduces with time (raw 158). But, in the figure the amplitude is zero in the range of 0-9 s, then it increases in the range of 10-14 s, and then it decreases. What really shown in the figure"?*

A22: A confusing figure was used. It has been deleted.

**Summary of changes**

**(Reviewer #2)**

First, we would like to thank the reviewer for his interest in our work and for helpful comments that will drastically improve the paper. As indicated below, we have checked all comments provided by the reviewer and have addressed necessary changes accordingly to his feedback.

C1: *"The paper does a poor job of placing the work in a context with previous work that relates fault slip behavior to elastic oscillations of the rock surrounding the fault. Addressing this comment will make the paper more readable to a wide earth science audience and place it in better context to other work that has been done on a similar topic".*

A1:  Thank you for your suggestion. The additional literature review part has been added.

C2: *"An application of simple models like the Burridge-Knopoff model and 1D model of an infinite elastic rod driven by elastic shear spring for the declared purpose should be substantiated in details".*

A2: The original BK model consists of an assembly of blocks, where each block is connected via the elastic springs to the next block and to the moving plate. In the present paper, we simulate the simple one-dimensional version of BK model, which consists from one block.

Additional details and description of these models were added into the paper.

*C3: "The constant friction factor used in the models instead of generally accepted rate-and-state friction law has to be grounded and supported by lab results and field observations".*

A3: We do not advocate constant friction. We just demonstrated that even with constant friction a stick-slip like behaviour is possible. We now added discussion where we analyse the effect of rate-dependent friction.

*C4: "A discussion section of the manuscript is required for an analysis and comparison of the numerical results and drawn conclusions with published data obtained under laboratory and natural conditions".*

A4: We agree with the reviewer. The discussion part has been added.

*C5: "Moreover, I realized that the English writing is not good enough, some parts of the text are difficult for understanding, there are some syntax and spelling errors, and I strongly recommend reviewing the text by a native English speaker".*

A5: Thank you for your suggestion. This has been done.

**List of all relevant changes**

1. Discussion part has been added
2. Table 1 has been added
3. The text was modified: "It is observed that despite the presence of shear springs and friction between the rod and the stiff surface, the waves propagate with the p-wave velocity determined by the Young's modulus and density of the rod."
4. The captions in the paper have been modified
5. Raw 38. Cohee and Beroza, 1994a → Cohee and Beroza, 1994" has been modified

6. *"However, the faults ... can produce sliding over initially stable fractures/interfaces" – a citation is needed".* Citation has been added
7. *"Raw 64. The citations should better be replaced by (Brace & Byerlee, 1966)".* The citation has been replaced.
8. Equations 1-4 have been corrected.
9. Figure 2 has been replaced.
10. Additional para has been added: "Furthermore, the energy in the system does not change with time, obviously due to the constant energy influx by velocity $V_0$, where the excess of the $V_0$ is dissipated by friction".
11. *"Raw 114. $\tau fr = k\mu\sigma N$. What is $k$"?* Formula has been corrected.
12. Equation 6 has been corrected
13. Awkward notations were used in eq. 8-14. *U* was not to be displacement. It has been changed.
14. The paper structure and data presentation have been modified.
15. Figures 1, 2, 3, 4 have been modified.
16. A standard mathematical formula was added, please see equation 15.
17. Additional literature review has been added.
18. The references part has been modified.
19. Additional details and description of present models have been added into paper.

[revised manuscript text omitted]